# An Immunological Perspective on the Mechanism of Drug Induced Liver Injury: Focused on Drugs for Treatment of Hepatocellular Carcinoma and Liver Transplantation

**DOI:** 10.3390/ijms24055002

**Published:** 2023-03-05

**Authors:** Soon Kyu Lee, Jong Young Choi, Eun Sun Jung, Jung Hyun Kwon, Jeong Won Jang, Si Hyun Bae, Seung Kew Yoon

**Affiliations:** 1Division of Hepatology, Department of Internal Medicine, Incheon St. Mary’s Hospital, College of Medicine, The Catholic University of Korea, Seoul 06591, Republic of Korea; 2The Catholic University Liver Research Center, College of Medicine, The Catholic University of Korea, Seoul 06591, Republic of Korea; 3Division of Hepatology, Department of Internal Medicine, Seoul St. Mary’s Hospital, College of Medicine, The Catholic University of Korea, Seoul 06591, Republic of Korea; 4Department of Pathology, Eunpyeong St. Mary’s Hospital, College of Medicine, The Catholic University of Korea, Seoul 06591, Republic of Korea; 5Division of Hepatology, Department of Internal Medicine, Eunpyeong St. Mary’s Hospital, College of Medicine, The Catholic University of Korea, Seoul 06591, Republic of Korea

**Keywords:** hepatocellular carcinoma, hepatotoxicity, immune checkpoint inhibitors, injury, tyrosine kinase inhibitors, immunosuppressant, microenvironment, T cell, B cell, macrophage

## Abstract

The liver is frequently exposed to potentially toxic materials, and it is the primary site of clearance of foreign agents, along with many innate and adaptive immune cells. Subsequently, drug induced liver injury (DILI), which is caused by medications, herbs, and dietary supplements, often occurs and has become an important issue in liver diseases. Reactive metabolites or drug–protein complexes induce DILI via the activation of various innate and adaptive immune cells. There has been a revolutionary development of treatment drugs for hepatocellular carcinoma (HCC) and liver transplantation (LT), including immune checkpoint inhibitors (ICIs), that show high efficacy in patients with advanced HCC. Along with the high efficacy of novel drugs, DILI has become a pivotal issue in the use of new drugs, including ICIs. This review demonstrates the immunological mechanism of DILI, including the innate and adaptive immune systems. Moreover, it aims to provide drug treatment targets, describe the mechanisms of DILI, and detail the management of DILI caused by drugs for HCC and LT.

## 1. Introduction

Drug-induced liver injury (DILI), an injury to the liver or biliary system caused by medications, herbs, or dietary supplements, accounts for 50% of acute liver failure cases in the United States [1,2]. DILI is classified as intrinsic (or direct) or idiosyncratic according to its pathogenesis [3]. Intrinsic DILI, which is predictable and acute-onset, occurs in a dose-dependent manner and can be reproduced in animal models [2,4]. However, idiosyncratic DILI, the most frequent type, is unpredictable and not dose-related DILI, although a minimum dose of 50 mg/day is usually required for its development [5].

The incidence of DILI varies by study design and cohort. Retrospective cohorts show lower incidence rates of DILI than prospective studies. According to several prospective studies, the annual incidence of DILI is approximately 13.9–19.1 per 100,000 inhabitants [6,7]. DILI can be influenced by multiple factors, such as age, sex, environmental exposure, and genetics, including human leukocyte antigen (HLA) [8,9]. Its diagnosis is based on an appropriate temporal relationship between drug intake and liver injury, along with the exclusion of other possible causes of liver damage, including viral infection and alcohol consumption [2]. The Roussel Uclaf Causality Assessment Method (RUCAM) is the most widely used assessment scale for DILI [10]. Moreover, according to elevated liver enzyme levels, represented as the alanine aminotransferase (ALT)/alkaline phosphatase (ALP) ratio (R), DILI patterns can be determined as follows: hepatocellular pattern (R ≥ 5), cholestatic pattern (R ≤ 2), and mixed pattern (2 > R < 5) [2,11]. Recently, the updated RUCAM of 2016 was introduced to improve the diagnostic accuracy of DILI [12]. According to the updated RUCAM, assessment of DILI is differently suggested according to the pattern of DILI using ALT/ALP ratio (R) at first presentation. The updated RUCAM also presents a check list of differential diagnosis of DILI and criteria for a positive result of DILI following unintentional re-exposure [12]. The diagnosis of DILI can be confounded by several factors, including comedication and concomitant diseases; therefore, causality assessment using the updated RUCAM is important.

Recent studies have suggested that specific human leukocyte antigen (HLA) genotypes, such as HLA-B*5701, are risk factors for the development of DILI in patients receiving some drugs [13,14]. However, HLA genotypes cannot sufficiently explain the risk of DILI. Moreover, microsomal cytochrome P450 (CYP) also play a role in the development of DILI [14]. As CYP is involved in the metabolism of many drugs, various isoforms of CYP, including CYP3A4, may be associated with the development of DILI [15]. Population-based studies have also demonstrated that pre-existing liver disease, concomitant severe skin reactions, and comedications, such as nonsteroidal anti-inflammatory drugs, are associated with the development DILI [6,16]. Furthermore, ferroptosis can also be a potential factor in the pathogenesis of DILI [17]. Ferroptosis, an iron-dependent form of cell death, reduces cystine uptake causing the production of lethal reactive oxygen species, which can lead to the development of DILI [18].

Regarding immunologic perspective, the liver is the primary site of the clearance of foreign chemical agents; thus, it is exposed to many potentially toxic chemicals that can cause hepatocyte damage via mitochondrial dysfunction and oxidative stress [19]. In addition, the liver is an immune organ with abundant innate (e.g., neutrophils, natural killer [NK] cells, and Kupffer cells) and adaptive (T cells and B cells) immune cells [20]. Although the liver is an immunologically tolerant organ, immune responses, including innate and adaptive immune cells, play pivotal roles in the development of DILI. Tyrosine kinase inhibitors (TKIs), such as sorafenib, lenvatinib, and regorafenib, were developed to treat advanced hepatocellular carcinoma (HCC). Moreover, recent studies have demonstrated the high efficacy of immune checkpoint inhibitors (ICIs), including atezolizumab plus bevacizumab, in HCC [21,22]. Along with the high efficacy of these novel drugs, DILI has become a critical issue in ICI use.

In this review, we discuss the immunological perspective of the mechanism of DILI, including the innate and adaptive immune systems (Figure 1). Moreover, we describe the frequency, hepatobiliary manifestations, and mechanism of DILI in patients with HCC treated with TKIs and ICIs. We also demonstrate the development of DILI in liver transplant (LT) patients administered immunosuppressants (ISs).

## 2. Immunological Perspective on DILI Mechanism

### 2.1. Danger Hypothesis

T cell-mediated liver injury is the cornerstone of DILI development [23]. The hapten hypothesis, which suggests that haptens make the proteins “foreign” and lead to their recognition and destruction by the immune system, was introduced to explain this immune response [24]. However, this hypothesis is insufficient to support the strong immune response in DILI. Subsequently, the danger hypothesis was proposed to redeem the hapten hypothesis. The generation of reactive metabolites or drug–protein complexes damages hepatocytes via several pathways, including oxidative stress, endoplasmic reticulum (ER) stress, bile salt export pump (BSEP) inhibition, and mitochondrial damage [3,25]. Damaged hepatocytes release several damage-associated molecular patterns (DAMPs), such as high-mobility group box (HMGB)-1, heat shock proteins, S100 proteins, and ATPs, which play a pivotal role in the activation of antigen-presenting cells (APCs) by producing a second signal (interaction of CD28 with B7 molecules) [26]. This co-stimulation often refers to a “danger signal” according to the danger hypothesis. Activated APCs lead to the activation of adaptive immune responses, including CD4^+^, CD8^+^, and B cells, which cause idiosyncratic DILI [26] (Figure 1).

### 2.2. Innate Immune Systems in DILI

As discussed above, reactive metabolites or drug–protein complexes can damage hepatocytes via ER and oxidative stress, inhibition of BSEP, and mitochondrial damage [3,25]. Damaged hepatocytes secrete DAMPs, including HMGB-1, heat shock proteins, S100 proteins, and ATPs, which activate the innate immune system and stimulate the immune response [27]. Activated innate immune systems (e.g., Kupffer cells, neutrophils, NK cells, and NK T cells) damage hepatocytes, recruit immune cells, and stimulate adaptive immune response during DILI (Figure 1) [28].

#### 2.2.1. Kupffer Cells

Kupffer cells, resident macrophages in the liver, are important in DILI development. They play key roles in phagocytosis, antigen presentation, and pro-inflammatory cytokines [29]. Traditionally, Kupffer cells can be classified into two types as follows: M1, Kupffer cells that secrete pro-inflammatory cytokines, such as interleukin (IL)-6 and tumor necrosis factor alpha (TNF-α); M2, Kupffer cells secreting potent immunosuppressive cytokines [30,31]. During DILI, Kupffer cells are activated by DAMPs and release pro-inflammatory cytokines and reactive oxygen radicals, along with infiltrated macrophages [32]. Kupffer cells also produce chemokine ligands to recruit monocyte-derived macrophages to the liver during the early phase of inflammation [33]. Activated Kupffer cells can exacerbate liver injury through these pathways.

#### 2.2.2. Neutrophils

Neutrophils, the first-line responders to bacterial and fungal infections, are the most abundant fraction of the innate immune cell group [34]. They defend against infection via phagocytosis, degranulation, and extracellular trapping [35]. Granulocyte colony-stimulating factor is a key regulator of neutrophil generation and maturation. The gut microbiome and metabolites may also play a role in neutrophil function [36]. During infection and inflammation, neutrophils are recruited to the site of inflammation via cytokine and chemokine production [34]. Neutrophils extravasate into the liver parenchyma via chemotactic signal from hepatocytes and other extravasated neutrophils. Extravasated neutrophils directly contact hepatocytes and trigger neutrophil activation. Eventually, abnormally activated neutrophils promote oxidative stress, mitochondrial dysfunction, and necrotic cell death, which can lead to acute liver injury during DILI [36]. Liver injury can be exacerbated by oxidative stress, involving myeloperoxidase and proteolytic enzymes [35,37].

#### 2.2.3. NK Cells

NK cells, the key players in liver immunity, are abundant in the liver, constituting 30–50% of intrahepatic lymphocytes [38]. NK cells have cytotoxic functions and express immunomodulatory cytokines, such as IL-1β, IL-2, IFN-γ, and TNF-α, which can be categorized into subsets according to their characteristics, including cytokines and cytotoxic capabilities [39,40]. These functions can also mediate DILI pathogenesis. The release of cytotoxic granzymes and perforin along with the production of TNF-α and IFN-γ can result in liver injury during DILI [41,42]. The IFN-γ production can mediate the infiltration of immune cells and release of cytokines, which results in hepatocyte apoptosis during DILI [42,43].

#### 2.2.4. NK T Cells

NK T (NKT) cells are unique lymphocytes that have both T and NK cell properties in their phenotype and function [44,45]. NKT cells, characterized by semi-variant T cell receptors (TCRs) and the major histocompatibility complex class I-like molecule CD1d, are pivotal in immunity against pathogens, bridging innate and acquired immunity [46,47,48]. These cells can be activated in both TCR-dependent and -independent manners and stimulate NKT cells to release cytokines, including IFN-γ and IL-17, which can recruit neutrophils, macrophages and activate adaptive immune responses, resulting in acute liver injury (DILI) [49,50]. However, studies have shown that NKT cells also have protective roles in liver injury and cancer immunology [28,51]. Recent studies have also demonstrated the potential role of the gut microbiome as a regulator of NKT cells, with further validation studies needed [51,52].

#### 2.2.5. Mast Cells

Mast cells (MCs) originate from hematopoietic stem cells and play a role in the initiating the response of the innate immune system [53,54]. MCDs are activated by DAMPs, cytokines, and chemokines [55,56,57]. Activated MCs undergo degranulation and release histamines and TNF, which activate the innate immune systems and exacerbate inflammation [58,59,60]. This process stimulates hepatic stellate cells, Kupffer cells, and pro-fibrogenic signaling pathways, which aggravate liver damage and fibrosis [61,62]. Recent studies have also demonstrated that activated MCs affect T cell activation and contribute to adaptive immunity [63,64].

### 2.3. Adaptive Immune Systems in DILI

The adaptive immune response is stimulated by activated innate immune systems, released DAMPs, and APCs presenting reactive metabolites or drug–protein complexes. The adaptive immune response, a critical process in acute injuries, includes CD4^+^ and CD8^+^ T-cell responses and B cell-mediated humoral reactions [65]. During DILI, activated CD4^+^ and CD8^+^ T cell and B cells damage hepatocytes. Meanwhile, regulatory T (Treg) cells and their functions are decreased, exacerbating liver injury in DILI [65] (Figure 1).

#### 2.3.1. CD4^+^ and CD8^+^ T Cells

Among T cells, CD4^+^ and CD8^+^ T cells are the main T lymphocytes in adaptive immune responses and are pivotal during liver injury [66]. The presentation of reactive metabolites or drug-protein complexes by APCs along with signal 2 activates CD4^+^ Th0 cells, which triggers a subsequent adaptive immune response [25,67]. Among subsets of CD4^+^ T cells, activated helper T (Th) 1 cells secrete IFN-γ, IL-2, and TNF-α and activate CD8^+^ T cells during DILI [68,69]. Th2 cells, an important subset of CD4^+^ T cells, release IL-4 and drive the proliferation and differentiation of B cells, which cause B cell-mediated humoral reactions [70,71]. Infiltrated CD8^+^ T cells, a major cell killer in adaptive immunity, have direct cytotoxic function and secrete granzymes, perforin, and cytokines, including TNF-α, IL-17 which cause cell death during DILI [65,72]. Indeed, infiltration of cytotoxic T cells (CTLs) may play an important role in fulminant drug-induced hepatic failure [73].

#### 2.3.2. B Cells

B cells originate from hematopoietic stem cells in the bone marrow. After maturation, B cells migrate from the peripheral blood into the spleen and germinal center [74]. As in other liver diseases, B cells participate in immune response and hepatocyte damage during DILI. B cells account for 8% of intrahepatic lymphocytes, which are activated and mature into plasma cells [75]. Plasma cells produce antibodies against proteins and damage hepatocytes during DILI [76]. During DILI, plasma cells can also produce autoantibodies against native proteins, such as cytochrome P450, which exacerbates liver injury [77].

#### 2.3.3. Treg Cells

Treg cells, accounting for 5–10% of CD4^+^ T cells, are crucial for maintaining immune homeostasis and tolerance in liver disease and transplantation [78,79,80]. Treg cells secrete IL-10 and TGF-β, suppressing the proliferation of CD4^+^ T and CD8^+^ T cells and secretion of IFN-γ [81]. Moreover, Treg cells inhibit the proliferation of Th17 cells and release of IL-17 [82]. A recent study demonstrated that Treg cells can be modulated by the gut microbiome in patients with autoimmune diseases, IBD, and transplantation, which might be associated with the pathogenesis of these diseases [83,84,85]. Indeed, a decrease in Treg cells induces an inflammatory response that leads to liver damage [86]. During DILI, intrahepatic Treg numbers and Foxp3 expression decrease, exacerbating liver injury with a decreased IL-10 level [87]. Increasing Treg cell numbers may alleviate liver injury via the secretion of IL-10 and TGF-β, which might be a treatment target for DILI [88,89].

## 3. DILI Caused by Drugs Treating HCC and LT

### 3.1. DILI Caused by Drugs Treating HCC

HCC remains a global burden, accounting for 800,000 deaths worldwide [90]. Despite the development of screening protocols and surgical or locoregional treatments for early HCC, diagnosis commonly occurs at the advanced stage [29]. Moreover, approximately half of all patients with HCC experience systemic therapies in their treatment history [91]. In the past decades, sorafenib, a TKI, has been used as the 1st line therapy for advanced HCC. Several TKIs, including lenvatinib, regorafenib, and cabozantinib, have been developed for the 1st and 2nd line treatment of advanced HCC [90]. Recently, immune checkpoint inhibitors, including atezolizumab plus bevacizumab, have shown high efficacy in the treatment of advanced HCC [21,22].

As described above, the liver contains various immune cell types, whose response to ICIs is mostly affected by the tumor microenvironment (TME), which is composed of Treg cells, tumor-associated macrophages (TAMs), cytotoxic T cells, myeloid-derived suppressive cells (MDSCs), and neutrophils [92,93]. The crosslinking between tumor cells and several immune cells causing an immuno-suppressive status has been a treatment target for ICIs to restore the immune response to HCC [94]. During ICI treatment, liver injury can be induced via direct or indirect immune pathways. In this section, we discuss the target, frequency, mechanism, and treatment of DILI caused by drugs for HCC (Table 1).

#### 3.1.1. Tyrosine Kinase Inhibitors

Several TKIs have been approved for treating advanced HCC (Table 1). Sorafenib targets vascular endothelial growth factor receptor (VEGFR), platelet-derived growth factor receptor (PDFGR), c-kit, and Raf, and it can inhibit cancer growth, progression, angiogenesis, and metastasis [95]. Lenvatinib is another multi-kinase inhibitor targeting VEGFR 1-3, PDFGR, fibroblast growth factor (FGF) receptors 1–3, RET, and KIT [96], and it showed non-inferior survival to and better progression-free survival than sorafenib [97]. Regorafenib, approved for HCC patients after sorafenib failure, also targets VEGFR 1-3, PDGFR, FGFR1-2, and RAF [98]. Cabozantinib has also been approved for sorafenib-experienced patients with HCC and targets VEGFR 1-3, MET, and RET [99]. These TKIs reinforce antitumor immunity by increasing dendritic cells (DCs), T-cell infiltration, and PD-1 expression on T cells. Moreover, TKIs also decrease pro-tumor immunity, such as a decrease in MDSCs, Treg cells, and M2 TAMs [100].

During treatment with TKIs, elevated serum aminotransferase levels are common (~50%); however, severe hepatitis with values greater than five times the upper limit of normal is rare [101]. However, several studies have reported that TKI-induced DILI is associated with progressive liver injury and failure [102,103]. Along with DILI, hand–foot syndrome and skin rash can be present in some patients who are administered TKIs, such as sorafenib and regorafenib [101,104,105]. In liver histopathology, hepatocellular necrosis is the most frequent manifestation of TKI-induced DILI, and immune-mediated hepatitis has also developed, including sorafenib-induced DILI [104]. Although the specific mechanism remains unclear, several TKIs, including sorafenib and regorafenib, are metabolized via the CYP 3A4 pathway, which may be associated with the production of a toxic intermediate (Figure 2) [101,105]. The direct effect of inhibition of cellular kinases, such as by lenvatinib and cabozantinib, can be another suggested mechanism for TKI-induced DILI [106,107]. TKI can also induce oxidative stress and apoptotic pathway activations, which can lead to immune response activation and TKI-induced DILI [104,108]. Moreover, several signal transduction pathways, including epidermal growth factor receptor and platelet-derived growth factor receptor, which interact with TKI, play pivotal roles in regulating DILI and are associated with TKI-induced DILI [109].

Owing to the possibility of DILI, the Food and Drug Administration recommends monitoring liver function with the use of some TKIs, including regorafenib. As TKI-induced DILI usually recovers its discontinuation, appropriate monitoring and dose reduction or temporary cessation can successfully control TKI-induced DILI [101,104].

#### 3.1.2. Immune Check Point Inhibitors

Recently, several ICIs have been approved for HCC treatment. Atezolizumab (an anti-PD-L1 antibody) plus bevacizumab (an anti-VEGF antibody) have changed the treatment landscape and paved the way for the combination therapy, with ICIs showing better overall survival than sorafenib [21]. Moreover, durvalumab (anti-PD-L1 antibody) and tremelimumab (anti-CTLA-4 antibody) also demonstrated superior survival rates compared with sorafenib [110]. In the second-line setting, pembrolizumab (anti-PD-1 antibody) monotherapy and nivolumab (anti-PD-1 antibody) plus ipilimumab (anti-CTLA4 antibody) have been approved for advanced-stage HCC [111,112] (Table 1).

The combination of anti-VEGF drugs with ICIs changes the tumor endothelium, increasing the infiltration of effector immune cells [113]. Moreover, combination therapy has a synergistic effect of increasing antitumor immune cell responses and inhibiting immunosuppressive pathways [114]. Indeed, ICIs that inhibit PD-1 or PD-L1 restore the function of effector CD8^+^ T cells [115]. CTLA-4 inhibitors activate naïve CD4^+^ and CD8^+^ T cells by promoting the interaction between costimulatory signals (B7 with CD28) [116]. Moreover, the addition of anti-VEGF drugs can show synergistic effects via several mechanisms, such as normalization of the vessel, which can lead to improvement in drug delivery and reduction in the immunomodulatory effect of VEGF on TAMs, MDSCs, Treg cells, and effector T cells [117].

ICI-induced DILI is an the immune-related adverse event, which is characterized by elevated aspartate aminotransferase (AST) and alanine aminotransferase (ALT) levels [117]. Although the pattern of ICI-induced DILI is heterogeneous, the hepatocellular type is usually frequent [118]. Using the RUCAM model, ICI-induced DILI usually begins 8–12 weeks after ICI initiation, although ICI-induced DILI can occur at any time [119,120]. The incidence of ICI-induced DILI is known to be higher in patients treated with combination therapy (up to 18%) than in those treated with monotherapy (up to 9%) [120,121]. Moreover, as patients with HCC usually have chronic hepatitis or cirrhosis, the incidence of ICI-induced DILI is more frequent than that in patients without liver cancer [122]. According to type and dose of ICIs, the incidence of ICI-induced DILI in any grade ranges from 8% to 20% and is the highest in patients treated with the combination of anti-PD-1 and anti-CTLA4 antibodies [111,123,124,125]. In the diagnosis of ICI-induced DILI, it is essential to exclude other confounding factors, including co-medication, concomitant diseases, and hepatic metastasis, as well as to evaluate the possibility of ICI-induced DILI based on RUCAM [15,126]. Moreover, ICI-induced DILI should be differentiated from autoimmune hepatitis (AIH) [127]. ICI-induced DILI usually has a negative or low titer of antinuclear and anti-smooth muscle antibodies and does not have a female preponderance [120].

Several mechanisms have been proposed to explain ICI-induced DILI development (Table 2 and Figure 2). The first is the reduction and depletion of Treg cells, which are essential immune cells for maintaining tolerance induced by ICI treatment, especially in CTLA-4 blockades [128,129]. The depletion of Treg cells subsequently induces the reduction of anti-inflammatory cytokines and proliferation of CD8^+^ T cells [130,131]. Moreover, early B cell changes, including elevation of the CD21^lo^ subtype, may induce autoreactive B cells, leading to ICI-induced DILI [132]. Representative histopathologic features of ICI-induced DILI are shown in Figure 2. Liver histopathology showed moderate portal inflammation with CD3^+^, CD4^+^, and CD8^+^ T cell infiltration along with periportal hepatocytic necrosis (Figure 3A–D). Predominant infiltration of histiocytes (CD68^+^ cells) was identified, along with mild infiltration of CD38^+^ cells, suggesting the presence of plasma cells (Figure 3E,F). ICI-induced DILI usually presents with lympho-histiocytic infiltration with lobular hepatitis, whereas AIH presents with interface hepatitis with plasma cell infiltration [120]. The gut microbiome may contribute to the development of immune-related adverse events (irAEs), especially immune-related colitis [133]. Gut microbial composition and their changes are associated with various liver disease and may influence the response to cancer immunotherapy [134,135,136]. In this context, the gut microbiota may be a biomarker for predicting irAEs including DILI. Further studies are needed to elucidate the specific pathogenic mechanisms underlying ICI-induced DILI.

ICI-induced DILI is asymptomatic in most cases; however, skin reactions (rashes) can occur in some patients [137]. Skin reactions are frequent irAEs after ICI treatment [138]. Moreover, irAEs frequently involve the gastrointestinal tract and endocrine organs, including the thyroid and lung [138]. The severity of ICI-induced DILI is classified according to the Common Terminology Criteria for Adverse Events (CTCAE) of the National Cancer Institute (Table 2) [139]. From grade 2, ICI-induced DILI is treated by stopping ICI along with corticosteroid [140,141]. In grade 2 DILI, 0.5–1 mg/kg/day of prednisolone is recommended, and in grades 3 and 4, the dose rises to 1–2 mg/kg/day of IV methylprednisolone [142]. High dose ursodeoxycholic acid (UDCA) can also be added for patients with cholestasis [118]. In patients with refractory to corticosteroids, mycophenolate mofetil (MMF), azathioprine, or tacrolimus have been used to improve liver function tests [142,143,144]. Although the time to resolution of ICI-induced DILI varies, patients with ICI-induced DILI usually recover within two weeks [145]. Reinduction of ICI after DILI can be applied to patients with grade 2 and 3 DILI, whereas patients with grade 4 DILI must permanently discontinue ICI [146]. Corticosteroids can increase the risk of bacterial infection; therefore, strict evaluation and diagnosis of ICI-induced DILI using the updated RUCAM are needed before the commencement of corticosteroid therapy [12,147]. Moreover, further studies are required to identify and validate predictors for ICI-induced DILI development.

### 3.2. DILI Casued by Drugs for Treating LT

#### Immunosuppressants

LT patients generally require life-long ISs due to the risk of graft rejection after LT [148,149]. The most used ISs are calcineurin inhibitors, mycophenolate mofetil (MMF), and the mammalian target of rapamycin inhibitors (mTORi) [150]. Of the calcineurin inhibitors, cyclosporine inhibits the activation of T cells by binding cyclophilin, whereas tacrolimus binds to intracellular proteins and inhibits calcineurin phosphatase activity [150]. Subsequently, the nuclear factors of activated T cells cannot move to the nucleus, which shuts down the production of IL-2, leading to a decrease in T-cell response [151]. MMF, another type of ISs, inhibits the formation of guanosine monophosphate by blocking inosine monophosphate dehydrogenase and suppressing T-cell proliferation [152,153]. The mechanism of action of mTORi, including sirolimus and everolimus, includes the inhibition of serine/threonine kinase activity, a family of phosphatidylinositol-3 kinases (PI3K), which inhibits the PI3K/Akt/mTOR signaling pathway, the transduction signal of IL-2 receptors, and T-cell proliferation [154,155] (Table 1).

Significant elevation of liver function, including AST and ALT, is not frequent with calcineurin inhibitors and mTORi [156,157]. Generally, the abnormalities in liver function tests caused by calcineurin inhibitors and mTORi are asymptomatic [158]. Mechanismically, calcineurin inhibitors and mTORi are mainly metabolized by the cytochrome P450 system (CYP 3A4), which may be associated with DILI. Liver injury can be caused by direct hepatotoxicity or activation of immune cells induced by its metabolites [156,157]. Only a small portion of patients receiving MMF treatment experience elevation in serum liver function [159]. MMF is not usually metabolized by cytochrome P450 enzymes, and MMF-induced DILI may be associated with mitochondrial damage and its immunogenic metabolites [160]. As IS-induced DILI is generally mild and self-limiting, dose reduction or pausing ISs can resolve DILI.

## 4. Conclusions

The liver contains many innate and adaptive immune cells and, during the development of DILI, reactive metabolites or drug–protein complexes initiate innate and adaptive immune responses, including neutrophils, Kupffer cells, NK cells, CD4^+^ T cells, CD8^+^ T cells, and B cells. Multiple activated immune cells damage hepatocytes, leading to DILI. Meanwhile, Treg cells and their functions are suppressed, exacerbating DILI. Understanding the underlying mechanism of DILI may provide clues for future treatment targets for DILI.

The TME, composed of Treg cells, TAMs, cytotoxic T cells, MDSCs, and neutrophils, affects HCC development and responses to TKIs and ICIs. Recently approved ICIs target PD-1/PD-L1 and CTLA-4 to restore the immune response in HCC. An activated immune response can cause irAEs, including DILI, via direct and indirect pathways. DILI caused by TKIs and ICIs is usually asymptomatic and recovers after drug discontinuation. ISs used in LT patients infrequently cause DILI and require regular tests to monitor of liver function. According to the degree of DILI, appropriate treatment with corticosteroids may be needed in severe cases. Along with advances in the treatment of HCC and LT, it is mandatory that future studies elucidate the specific mechanism and appropriate management of DILI.

## Figures and Tables

**Figure 1 ijms-24-05002-f001:**
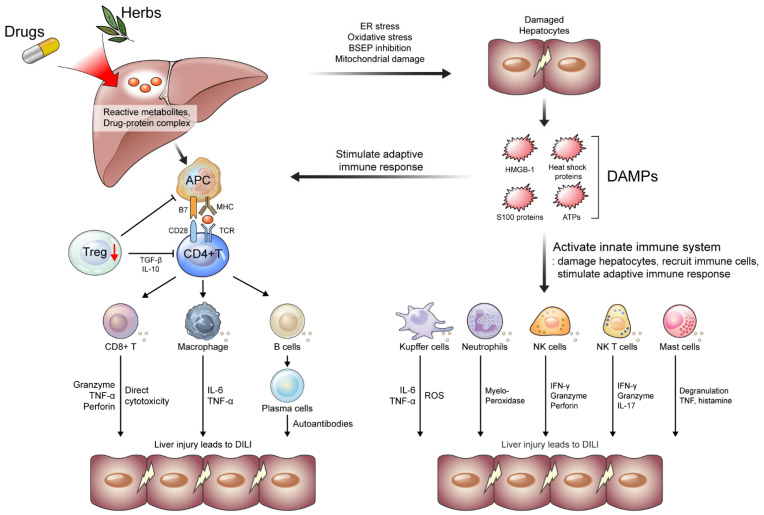
Mechanisms of the development of drug-induced liver injury (DILI). Reactive metabolites or drug–protein complex causes ER and oxidative stress in hepatocytes. BSEP inhibition and mitochondrial damage also damage hepatocytes, leading to the secretion of DAMPs, including HMGB-1, heat shock proteins, S100 proteins, and ATPs. DAMPs activate innate immune systems and stimulate immune response. Activated innate immune systems (e.g., Kupffer cells, neutrophils, NK cells, NK T cells, and Mast cells) damage hepatocytes, recruit immune cells, and stimulate adaptive immune response. Reactive metabolites or drug–protein complexes are presented by APCs, which lead to activation of adoptive immune response (e.g., T cells and B cells) along with the stimulation of APCs by DAMPs. Meanwhile, Treg cells decrease and fail to maintain immune tolerance. APC, antigen presenting cells; ATPs, adenosine triphosphate; BSEP, bile salt export pump; DAMP, damage-associated molecular patterns; ER, endoplasmic reticulum; HMGB, high-mobility group box; IFN, interferon; IL, interleukin; NK, natural killer; TNF, tumor necrosis factor; Treg, regulatory T cells.

**Figure 2 ijms-24-05002-f002:**
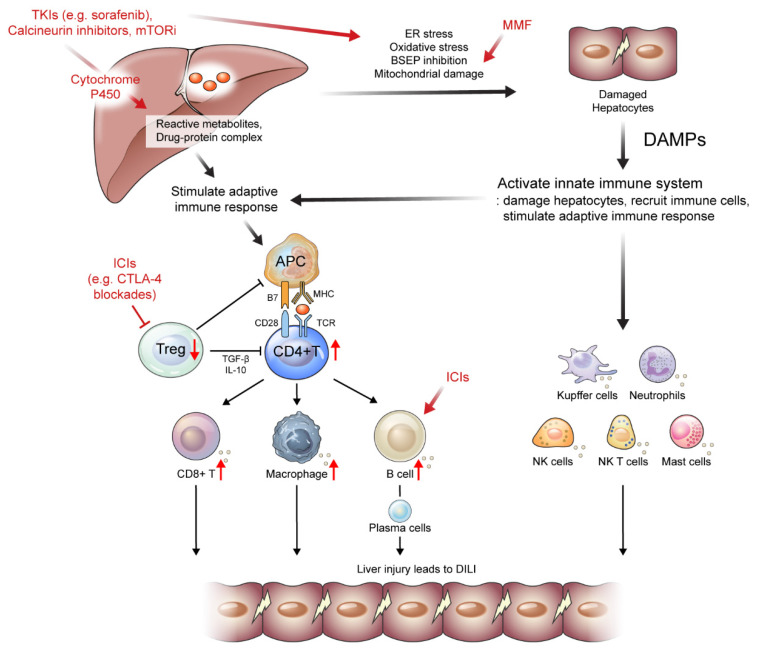
Suggested mechanisms of drug-induced liver injury (DILI) caused by drugs administered to patients with hepatocellular carcinoma or liver transplantation. Several tyrosine kinase inhibitors (TKIs), calcineurin inhibitors, and mTOR inhibitors are metabolized via the cytochrome P450 pathway, which may be associated with the production of a toxic intermediate. These drugs can also induce oxidative stress and apoptotic pathway activations, which can lead to the activation of immune response. Mycophenolate mofetil can induce mitochondrial damage, which then leads to DILI. Immune checkpoint inhibitors (ICIs) deplete Treg cells inducing the reduction of anti-inflammatory cytokines and proliferation of CD8^+^ T cells. Moreover, early B cell changes may induce autoreactive B cells, leading to ICI-induced DILI. APC, antigen presenting cells; BSEP, bile salt export pump; DAMP, damage-associated molecular patterns; ER, endoplasmic reticulum; ICIs, immune checkpoint inhibitors; MMF, mycophenolate mofetil; mTORi, mammalian target of rapamycin inhibitors; NK, natural killer; TKI, tyrosine kinase inhibitors; Treg, regulatory T cells;↑, an increase in the indicated cells; ↓, a decrease in the indicated cells; ┤, the reduction and depletion of indicated cells.

**Figure 3 ijms-24-05002-f003:**
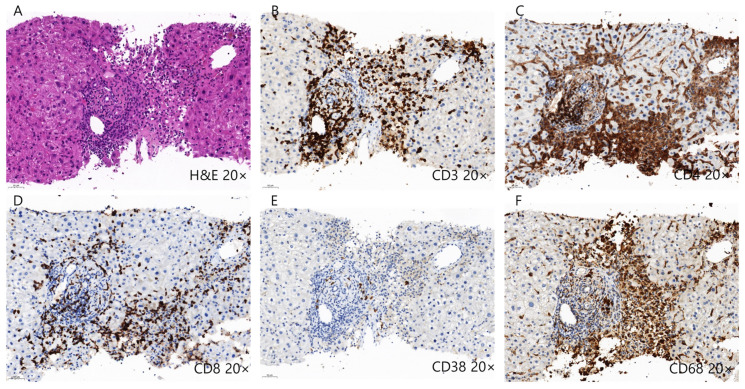
Histopathology of drug-induced liver injury induced by immune checkpoint inhibitors. (**A**–**D**) Liver histopathology shows moderate lymphocytic infiltration, including CD3^+^, CD4^+^, and CD8^+^ T cells along with periportal hepatocytic necrosis. (**E**,**F**) Predominant infiltration of CD68^+^ histiocytes were also identified along with mild infiltration of CD38^+^ plasma cells.

**Table 1 ijms-24-05002-t001:** Hepatobiliary manifestation and frequency of drug-induced liver injury caused by drugs for treating hepatocellular carcinoma and liver transplantation.

Drugs	Target for Drug Action	Hepatobiliary Manifestation *	Frequency
**Hepatocellular carcinoma—Tyrosine Kinase inhibitors**			
Sorafenib	Inhibits VEGFR, PDGFR, and Raf	R-value ≥ 5 (hepatocellular injury) LiverTox category: B	Common
Lenvatinib	Inhibits VEGFR, FGF, PDGFR, cKit, and RET proto-oncogene	2 < R < 5 (mixed liver injury) LiverTox category: D	Common
Regorafenib	Inhibits VEGFR, PDGF, RAF kinase, and c-Kit	R-value ≥ 5 (hepatocellular injury) LiverTox category: B	Common
Cabozantinib	Inhibits MET, VEGFR-2, and RET	2 < R < 5 (mixed liver injury) LiverTox category: E	uncommon
**Hepatocellular carcinoma—Immune checkpoint inhibitors and VEGF(R) inhibitors**			
Atezolizumab plus bevacizumab	Inhibits PD-L1, and VEGF	R-value ≥ 5 (hepatocellular injury) LiverTox category: B	14%
Durvalumab plus tremelimumab	Inhibits PD-L1, and CTLA-4	R-value ≥ 5 (hepatocellular injury) LiverTox category: B	20%
Nivolumab	Inhibits PD-1	R-value ≥ 5 (hepatocellular injury) LiverTox category: A	15%
Ramucirumab	Inhibits VEGFR-2	Infrequent liver injury LiverTox category: E	Rare
Nivolumab plus ipilimumab	Inhibits PD-1, and CTLA-4	R-value ≥ 5 (hepatocellular injury) LiverTox category: A	20%
**Liver transplantation—Immunosuppressants**			
Cyclosporine	Calcineurin inhibition	R-value ≤ 2 (cholestatic liver injury) LiverTox category: C	1–5%
Tacrolimus	Calcineurin inhibition	R-value ≥ 5 (hepatocellular injury) LiverTox category: C	5–10%
Sirolimus/Everolimus	mTOR inhibition	2 < R < 5 (mixed liver injury) LiverTox category: E	Rare
MMF	Antimetabolite (inhibit inosine monophosphate)	R-value ≥ 5 (hepatocellular injury) LiverTox category: D	Rare

* Hepatobiliary manifestations are demonstrated according to the R-value (ALT/ALP) of the RUCAM and LiverTox category. LiverTox category A is well known cause of immune mediated liver injury; category B, likely cause of clinically apparent liver injury; category C, probable rare cause of clinically apparent liver injury; category D, possible cause of clinically apparent liver injury; category E, unproven but suspected rare cause of clinically apparent liver injury. CTLA-4, cytotoxic T-lymphocyte-associated protein 4; FGF, fibroblast growth factor; MET, hepatocyte growth factor receptor; mTOR, mammalian target of rapamycin; PD-1, programmed cell death 1; PDGF, platelet derived growth factor; PD-L1, programmed cell death ligand 1; R, ratio; RET, rearranged during transfection; VEGFR, vascular endothelial growth factors receptor.

**Table 2 ijms-24-05002-t002:** Mechanism and treatment of drug-induced liver injury caused by immune checkpoint inhibitor use.

**Immune Cells**	**Mechanism**		**Refs.**
Treg cells	Reduction in Treg cells and anti-inflammatory cytokines		[107,108]
Th1 cells	Increase in Th1 cells and pro-inflammatory cytokines causing activation of CTLs and macrophages		[107,108,109,110]
CTLs	Stimulate proliferation of CD8^+^ T cells		[107,108,109,110]
B cells	Ealy B cell changes including the elevation of CD21^lo^ subtype may induce the autoreactive B cells		[111]
**Grade of DILI**	**Definition**	**Management**	[116]
Grade 1	Asymptomatic, T.bil > 1.5×ULN, AST or ALT > 1–3×ULN	Monitoring, continue ICI	[116,117,118,119]
Grade 2	Asymptomatic, T.bil > 1.5–3×ULN, AST or ALT > 3–5×ULN	Discontinue ICI, start 0.5–1.0 mg/kg/day of prednisolone with a taper, consider restart after recovering from DILI	[116,117,118,119]
Grade 3	Symptomatic, Fibrosis, Compensated cirrhosis, T.bil > 3–10×ULN, AST or ALT > 5–20×ULN	Discontinue ICI, 1–2 mg/kg/day of IV methylprednisolone with a taper, consider liver biopsy, consider restart after recovering from DILI	[116,117,118,119]
Grade 4	Decompensated symptom (ascites, encephalopathy, coagulopathy), T.bil > 10×ULN, AST or ALT > 20×ULN	Permanently discontinue ICI, 1–2 mg/kg/of IV methylprednisolone with a taper, consider liver biopsy	[116,117,118,119]
Grade 5	Death due to DILI		[116]

AST, aspartate transaminase; ALT, alanine transaminase; CTLs, cytotoxic T lymphocytes; DILI, drug induced liver injury; ICI, immune checkpoint inhibitors; T.bil, total bilirubin; Th, helper T; Treg, regulatory T; ULN, upper limit of normal.

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
