# Peer review of "An Immunological Perspective on the Mechanism of Drug Induced Liver Injury: Focused on Drugs for Treatment of Hepatocellular Carcinoma and Liver Transplantation"

_ijms, 2023, doi:10.3390/ijms24055002_

Round 1

Reviewer 1 Report (Previous Reviewer 2)

The authors have satisfactorily addressed most of my concerns and made the necessary changes to the manuscript.

Reviewer 2 Report (Previous Reviewer 3)

Dear authors 

The authors kindly submitted the revised version of the review manuscript entitled "An immunological perspective on the mechanism of drug-induced liver injury: Focused on drugs for treatment of hepatocellular carcinoma and liver transplantation." The authors comprehensively reviewed the immunological mechanism and drugs treating HCC and liver transplantation.  The authors revised the draft according to the reviewers' comments. After reviewing the revised manuscript, no further comments were given. 

Reviewer 3 Report (Previous Reviewer 1)

Manuscript is now sufficiently revised and should be accepted in the current form.

This manuscript is a resubmission of an earlier submission. The following is a list of the peer review reports and author responses from that submission.

Round 1

Reviewer 1 Report

Its a well organized review explaining the involvement of immune cells in DILI. Authors should add references as a separate column in table 2. Apart from this, I do not have any other comments and would like to appreciate the efforts of the authors for putting this nice piece of work.

Reviewer 2 Report

This is a well-written paper by Soon Kyu Lee and colleagues summarizing the main milestones in the field of drug induced liver injury in patients undergoing the HCC treatment or liver transplantation. Authors collected and reviewed recent literature evidence on innate and adaptive immune systems and use of tyrosine kinase inhibitors / immune check point inhibitors in the context of developing  drug induced liver injury - DILI.  Review is organized as follows: 1. Introduction; 2. Immunological perspective on DILI mechanism; 3. DILI caused by drugs treating HCC and LT (DILI caused by drugs treating HCC, DILI casued by drugs for treating LT); 4. Concluding remarks. Accordingly, the authors provide the evidence that TME, composed of Treg cells, TAMs, cytotoxic T cells, MDSCs, and neutrophils, can affect HCC development and responses to TKIs and ICIs. In turn, activated immune cells by ICIs response can cause DILI, via direct and indirect pathways. Although, DILI caused by TKIs and ICIs or immunosuppresants is usually asymptomatic and may recover after drug discontinuation it needs regular tests to monitor liver function. Finally, severe DILI may require appropriate treatment with corticosteroids. Authors extensively discussed latest literature data. Manuscript is illustrated with 1 figure explaining „Mechanisms of the development of drug-induced liver injury (DILI). Along with figure two tables are presented: Table 1. Hepatobiliary manifestation and frequency of drug-induced liver injury caused by drugs for treating hepatocellular carcinoma and liver transplantation; Table 2. Mechanism and treatment of drug-induced liver injury caused by immune checkpoint inhibitor use. Paper has 136 references, including few self-citations which are relevant to article's subject.

This is an interesting review-study, and clinically valuable, especially for those clinicists who use TKI/ICI modalities. This manuscript provides basic information on this issue.

Major comment:

Comment 1. It might be beneficial for readers to illustrate how representative TKIs/ICIs/ their metabolites initiate, or exacerbate DILI at the molecular level, especially in patients undergoing the HCC treatment or liver transplantation.

Minor comments:

Comment 1. I am curious whether there are individual predisposition for developing DILI, e.g. genetic conditions, concomitant diseases. If so, such information would also fit into the paper.

Comment 2. Text line 246-250 could be rephrased as it is hard to follow.

“These TKIs contribute to increased antitumor immunity by increasing dendritic cells (DCs), T-cell infiltration, and PD-1 expression on T cells, whereas TKIs decrease pro-tumor immunity, including a decrease in MDSCs, Treg cells (lenvatinib), and M2 TAMs (sorafenib) [82].”

Comment 3. Text line 299, D is missing “The incidence of ICI-induced DILI is known to be higher in patients treated with combination therapy than in those treated with monotherapy.

 Taken together, paper by Soon Kyu Lee and colleagues represents a worthwhile contribution to the DILI treating research. I recommend the manuscript for further publication process.

Reviewer 3 Report

Dear authors

The authors kindly submitted a review article entitled, An immunological perspective on the mechanism of drug induced liver injury (DILI): Focused on drugs for treatment of hepatocellular carcinoma and liver transplantation. Generally, the immunological viewpoints on the mechanism of DILI, including the innate and adaptive immune systems, is discussed. In addition, the frequency, hepatobiliary symptoms, and mechanism of DILI in HCC patients treated with TKIs and ICIs and immunosuppressants in liver transplant recipients are discussed. 

It is well written and an informative and valuable summary that does not show any significant bias. The authors have reviewed several important articles with organized writings. Generally, it needs only minor revisions before being accepted. The followings are my comments for authors:

1. In Figure 1, plasma cells might need to be mentioned with B cells, although they are the mature type of B cells and produce antibodies. 

2. The mast cells are not discussed in this review. However, mast cells were discussed in some articles associated. Please try to complete the part.

3. The authors did not clearly discuss the drug allergy which sometimes overlaps with DILI. Which drugs and in which types of allergy reaction may need to be discussed. 

4. DILI with skin reactions is not uncommon. It may be the occurrence of  general immune responses and a sign/symptom of DILI. The authors may think about its discussion.